# Barriers and facilitators to self-care practices for sexual and reproductive health among women of reproductive age

Amit Timilsina[1,2]*, Buna Bhandari[3,4], Alexandra Johns[2], Subash Thapa[5]

1 Ipas Nepal, Kathmandu, Nepal, 2 Asia Pacific Alliance for Sexual and Reproductive Health and Rights, Bangkok, Thailand, 3 Department of Global Health and Population, Harvard T.H Chan School of Public Health, Boston, Massachusetts, United States of America, 4 Central Department of Public Health, Institute of Medicine, Tribhuvan University, Kirtipur, Nepal, 5 Rural Health Research Institute, Charles Sturt University, Orange, New South Wales, Australia

* timilsinaamit@gmail.com

## Abstract

### Introduction

Sexual and Reproductive Health and Rights (SRHR) have been promoted globally, yet sexual and reproductive health (SRH) interventions are seldom evaluated from the perspective of service users and service providers. Very little is known about whether and why various target groups including general women are (or are not) practicing SRH -related self-care practices. This study explored SRH self-care practices and facilitators and barriers to the adoption of SRH self-care among reproductive-age women of Nepal.

### Methods

In this descriptive qualitative study, we conducted in-depth interviews in June 2022 with ten married women of reproductive age (service users) and four SRHR service providers (program managers and health service providers) in Nepal. Thematic analysis was conducted for data analysis.

### Results

We found that commonly practiced self-care practices were self-administration of contraceptives, self-management of pain, self-monitoring of pregnancy, self-awareness and seeking medical abortions (tele-abortion), self-medication for pre-exposure prophylaxis for HIV, and self-testing for HIV and pregnancy. The multi-level barriers to SRH self-care were poor knowledge and perceived lack of need for SRH self-care, limited access, and negative behaviors from the service providers. The program-related barriers included lack of evidence, limited financial resources, lack of accountability, and limited knowledge and skills among service providers on SRH self-care measures. Peer support, an increasing number of service sites, and access to and use of digital (health) tools emerged as the facilitators of SRH self-care.

**Data Availability Statement:** The codebook produced as the outcome of the data analysis, the interview guideline for the IDI and KII, and the informed consent forms have been available for

this study. As participants have not consented to the availability of full interview transcripts during this study, it has not publicly available to respect participants' anonymity and consent. However, if any additional data is required, it can be made available upon direct request to the data repository at info@asiapacificalliance.org respecting the confidentiality and anonymity of the participants.

**Funding:** The author(s) received no specific funding for this work.

**Competing interests:** The authors declare no conflicting interests exists.

## Conclusions

The findings of this study suggest that addressing barriers such as poor knowledge, limited access, and negative attitudes while leveraging facilitators such as peer support and digital tools is essential for promoting and enabling effective SRH self-care among women. Population-wide awareness programs supplemented by increasing service sites are essential for increasing SRH self-care practices.

## Introduction

The World Health Organization (WHO) recommends a range of self-care interventions, such as self-injection (e.g., of contraceptives), self-screening or testing (e.g., for sexually transmitted infections and pregnancy), self-medication (e.g., for abortion and HIV), and self-monitoring (e.g., of fertility), which should be delivered to the individuals as a human right in the context of national health systems [1,2]. Some of these interventions are self-administered without healthcare providers being involved, and some are integrated within service provision.

Self-care for sexual and reproductive health (SRH) has the potential to support low and middle-income country (LMIC) governments in achieving a range of health-related international commitments and further contributes to a more robust and resilient health system that enables access to and empowerment of marginalized populations [3]. This includes a range of targets under the Agenda 2030 and the Sustainable Development Goals (such as 3, 5 and 10) along with those contained in global intergovernmental consensus documents, including the 1994 International Conference on Population and Development Programme of Action and the 1995 Beijing Platform for Action, the 2019 Political Declaration of the High-level Meeting on Universal Health Coverage, and the outcomes of their reviews [3]. The WHO highlights the need to better understand the ways self-care interventions can help to improve the health and well-being of individuals and how it empowers individuals and communities to better understand their rights, including SRHR [1].

Many national health systems globally have been promoting SRH self-care as integral to a "people-centered approach towards healthcare" [3–6]. During the COVID-19 pandemic and with the resultant widespread lockdowns, self-care interventions for SRH were acknowledged as important for enhancing access to SRHR, especially for women [3,7]. Some benefits on the individual level, as highlighted in a systematic review by Remme et al., are increases in self-testing for health problems and adoption of preventative practices and the use of SRH-related information and services and reduced non-direct health costs [8].

Global health programs, such as self-care interventions for SRH, can (and at times do) take on culturally distinctive significance in different local settings, and in some instances, there could be a tension between global policies and local reality that is foundational to public health practice [9]. A corollary of the social construction of reality is that each local world realizes values that amount to a local context or norms that influence the behavior of its members, which needs exploration. For instance, certain self-care interventions may require guidance and additional support from the health system and service providers to be effective and appropriate for target groups. Service providers can explain the local understanding and needs of the women and girls as ways to provide support [1].

The available evidence also highlights the fact that self-care needs to be looked upon more contextually and critically as self-care practices surrounding HIV testing and abortions are largely influenced by local contexts and restrictive norms and policies [4,10–13]. The available

studies on self-care interventions reflect increasing evidence regarding self-care interventions, particularly on HIV self-testing and HPV self-sampling, but very little evidence exists on abortion, contraception and family planning, digital self-care and general SRH self-care [4,6,14]. In South Asian countries, the need for more evidence to understand current self-care practices is highlighted by the toolkit developed by Asia Pacific Alliance for Sexual and Reproductive Health and Rights in the COVID-19 context. How self-care measures are adopted and practiced by women is definitely one of the least investigated areas of selfcare [3].

In LMICs such as Nepal, SRH interventions for reproductive age women mainly include the provision of education and information, and counseling services and health services, such as contraceptive and safe abortion services, HIV prevention programs including mainly community-based testing and HIV self-testing [15]. The pilot testing of a self-injectable hormone for contraception revealed a higher continuation rate, while a pilot tele-abortion program during COVID-19 was found to face implementation problems and was then discontinued [15,16]. Thus, the barriers for selfcare could vary according to the service types, but there are also more general program-related factors such as poor implementation and other social and cultural factors that limit target groups' ability to access them.

Evidence on selfcare practices for SRH, and social- as well as program-related barriers in adopting such practices is necessary to understand and design appropriate program approaches that ensure SRH self-care is accessible and practiced with dignity and free from stigma and shame. However, so far, none of the studies have explored SRH related self-care practices, and barriers in adopting the practices particularly from the perspectives of the service providers and service users in the Nepalese context. This study explored SRH self-care practices and facilitators and barriers for the adoption of SRH self-care among reproductive age women.

## Methods

### Study design and settings

This was a descriptive qualitative study conducted in Kathmandu, Nepal, a locale having situated most of the organizations actively working in the field. This study was approved by the ethical board of Nepal Health Research Council (reg no. 3093) and informed written consent was taken from all the participants except for the women with disability for whom verbal consent was taken.

### Participants

The study participants primarily included service providers and target groups of self-care interventions. The service providers were mainly program managers who have been actively working in the SRHR field and health service providers providing SRH self-care services. The target groups included married women (pregnant or with at least one child) from the general population who were at least 18 years of age and using SRH self-care services. The participants who did not provide consent for participation were not included in the study.

### Sampling and recruitment

Purposive sampling was used to select the participants. A prioritized list of target groups who were willing to participate in the interview and able to better explain their experiences was made in consultation with the health care provider and community workers. For the target group of SRH selfcare interventions, we consulted with the community program managers and invited 10 married women who were either pregnant or had a child. Married women who were pregnant or had children were chosen to ensure better understanding and practice of

SRH self-care. In total, 2 health service providers and 2 community workers were interviewed in June 2022 to collect the data and information regarding SRH self-care practice from service providers who provide family planning, safe abortion, HIV-related self-care services. These program managers and health service providers were directly approached in their offices for an appointment, oriented about the research and its importance and asked for their written consent to participate in this study. Two participants from each of the program managers' group, and the health service providers' group agreed to participate in the study. All the participants of this study from the list were contacted via phone and asked for their verbal consent to participate in the study in the call, and out of 15 participants identifying themselves as women, only 10 of the women participants agreed to participate in the study. The Nepal Health Research Council (Ref. No: 3093) reviewed the research proposal and provided ethical clearance for and approved the study. The research follows standard ethical principles guided by the Declaration of Helsinki (ethical principles. The autonomy and confidentiality of participants have been maintained throughout the project period. All the participants were oriented about the study and its importance. Prior to data collection, written consent was taken from the participants after orientation, verbal consent was taken before starting the interviews and their right to withdraw from the research was reiterated. The participants were assured of anonymity and confidentiality before, during and after the study.

## Data collection

A semi-structured interview guide was developed including topics such as previous SRH self-care practices, reasons for practicing, perceptions about how community members think about/use SRH self-care practices, potential barriers and facilitators, and knowledge about SRH self-care interventions and organizations. To check whether the questions were appropriate to the local context, understandable and relevant, the interview guide was pretested with two women and some questions were edited for clarity and omitted from the final version of the interview guide. The data from the two pretested participants were not included in the final analysis.

Initially, we interviewed service providers and then continued our interviews with the target groups of different backgrounds. Based on participants preferences; four interviews were conducted online through Zoom, and the rest were conducted face-to-face. Having a mix of interviewee backgrounds not only increases the diversity of perspectives and experiences but can also help in developing a more comprehensive understanding of the phenomenon being studied, thereby aiding in achieving data saturation [17]. Furthermore, we believe that face-to-face interviews allow for in-depth exploration of the topic and non-verbal cues, while online methods provide a convenient and accessible way for participants who are geographically dispersed or have time constraints to participate. Using both methods can help reach a diverse group of participants and increase the amount of data collected.

Altogether, we conducted ten interviews, and prior to each interview, verbal consent was taken and tape-recorded. All interviews were conducted privately, and on average, they took 35–45 minutes. The socio-demographic characteristics of the participants are summarized in Table 1.

## Data analysis

The data for analysis comprised of transcripts of audio recordings from the in-depth interviews. The first author transcribed all the recordings in Nepali, which were then translated into English. Another researcher checked the quality of translations, and necessary reconciliations were made after discussions. For data analysis, we opted for a thematic approach,

**Table 1. Socio-demographic characteristics of the study participants.**

| Participants | Type of participant | Age | Background | Highest academic degree | Marital status | Employment status |
|---|---|---|---|---|---|---|
| I1PW34 | Service user | 34 | Female (Pregnant woman) | Masters | Married | Employed |
| I2PW30 | Service user | 30 | Female (Pregnant woman) | Masters | Married | Employed |
| I3GW33 | Service user of abortion and family planning services | 33 | Female (General woman) | Masters | Married | Employed |
| I4GW27 | Service user of family planning services | 27 | Female (General woman) | Masters | Married | Employed |
| I5PW30 | Service user of family planning and digital app | 30 | Female (Pregnant woman) | Masters | Married | Employed |
| I6PW31 | Service user of family planning, pregnancy kit, digital app | 31 | Female (Pregnant woman) | Masters | Married | Employed |
| I7GW31 | Service user of family planning, pregnancy kit, digital app | 31 | Female (General woman) | Masters | Married | Employed |
| I8GW28 | Service user of family planning, pregnancy kit, digital app | 28 | Female (General woman) | Bachelors | Married | Unemployed |
| I9GWD25 | Service user of family planning, pregnancy kit, digital sources | 25 | Female (With visual impairment) | High school | Married | Unemployed |
| I10GWD32 | Service user of family planning, digital sources | 32 | Female (With visual impairment) | Bachelor | Married | Employed |
| K11HW54 | Health care provider (family planning and safe abortion) | 54 | Female | Masters | Married | Employed |
| K12HW27 | Health Care Provider for HIV and AIDS | | Male | Masters | Married | Employed |
| K13CW27 | Community worker for Family Planning | 33 | Male | Masters | Unmarried | Employed |
| K14CW35 | Community worker on HIV and for sexual and gender minorities | 35 | Transwoman | Bachelor | Unmarried | Employed |

proposed by Braun and Clarke, which allowed us to examine the perspective of research participants with diverse backgrounds to examine similar and contrasting perceptions that could help to provide detailed understanding of the issues [18]. Nvivo-10 software was used to support the analysis.

At first, initial codes or themes were developed inductively by examining the data. This process included thoroughly reading the transcripts and labelling the codes in each line expressing a concept related to the phenomenon under study, which resulted in over 70 initial codes (see Table 2). Two rounds of discussion were organized between researchers to discuss the codes and one discussion was held after the coding process to review and refine the codes (Table 2). Themes and sub-themes were developed from the coded data such as digital health, implementation, mental health, maternal health, selection of family planning methods, HIV-testing, ART services, societal norms and culture, family support, unmet needs, knowledge gap, implications, information, benefit, policy and guidelines, personal behavior etc.

The conceptual themes and sub-themes that emerged from the data were created from the initial codes based on repeating ideas that were similar in meaning and the relationships that appeared between the codes. We looked for patterns across all the data to understand the phenomena. This process involved the description and interpretation of data with a focus on participants' accounts of their beliefs, experiences, and the meanings they attributed to them.

## Results

The study presents the three major themes (SRH self-care practices among the women, Barriers for adopting self-care practice for SRH, Facilitators of adopting SRH services) and 12 sub-themes emerged which includes Self-testing, Self-awareness, Self-management, Poor knowledge, perception and stigma regarding SRH self-care practices, Limited access to self-care SRH services, Negative attitude of service providers, Program related barriers, Information and

**Table 2. Code tree obtained from the data analysis.**

| Theme | Sub-theme | Codes |
|---|---|---|
| Current SRH self-care practices | Self-testing | Self-testing for pregnancy, HIV self-testing |
| | Self-awareness | Self-monitoring for pregnancy (tracking ANC dates and health complications); Self-medication for pain management; Healthy food consuming (e.g., Avoiding specific food- junk foods, coffee, tobacco) and physical activity (e.g., Yoga for mental well-being); Self-awareness about and personal hygiene |
| | Self-management | Use of medical abortion services and tele-abortion services, Self-medication in the form of Pre-Exposure Prophylaxis for HIV (PrEP); Use of condoms to prevent HIV; use of contraceptive pills and condoms; Self-management of menstrual problems (Taking pain killers for pain; tracking menstrual period; taking rest/drinking soups/using hot bags during cramps) |
| Barriers to self-care practices for SRH | Poor knowledge, perception and social stigma regarding SRH self-care practices | Behavior towards self-care measures, Limited knowledge and understanding regarding self-care measures; Regarding self-care measures; Societal norms and culture towards SRH self-care measures; Accessibility among marginalized women |
| | Limited access to SRH services | Access to toolkit, Adequacy of self-care services; implications of self-care measures, Limited SRH-self-care services; Limited SRH self-care options, Unmet need for self-care services; Dilemma to access SRH self-care measures; |
| | Negative attitude of service providers | Unwelcoming behavior and attitude of service; service provider bias; Influence against self-care measures |
| | Program related barriers | Limited financial means and resources; Poor accountability for self-care program; Lower awareness among service providers regarding self-care; Limited information regarding SRH self-care measures; Lower capacity of providers; lower individual capacity of women; poor satisfaction among participants regarding self-care measures |
| Facilitators of adopting selfcare practices for SRH | Information and knowledge regarding selfcare practices | General as well as need-based information; Counseling over phone for self-management of menstrual cramps; Telephone counseling for symptoms management, and information; Psychosocial support for medical abortion; Mental health support and increase tolerance among service providers; Confidential HIV (self) testing |
| | Family and Peer support | Initiation of self-care measures; help from peers; suggestions and feedback from peers; self-assessment of SRH self-care services; family support; trust towards SRH self-care measures; comfort while using SRH self-care measures |
| | Increasing number of NGO based service sites | advocacy on self-care interventions; availability of counseling services; evidence generation regarding SRH self-care interventions; opportunities for program and service expansion; policy and guidelines supporting SRH self-care program and services; policy gap regarding SRH self-care interventions; prioritization of self-care services and program; enabling environment for implementation of self-care services; SRH self-care selection choice of beneficiaries; HIV and AIDS interventions, family planning interventions; pregnancy related practice; self-testing, services offered by organizations; service strengthening strategies on self-care measures |
| | Use of digital (health) tools | availability of digital health services; services during COVID-19; menstruation related information and counseling; family planning services, information and counseling; communications and media |

knowledge regarding SRH self-care practice, Family and peer support, Increasing number of NGO based service sites, Use of digital (health) tool as depicted in Table 2.

## Self-care practices among the women

Our synthesis revealed target group differences in self-care practices and target group-specific unmet health needs. SRH self-care practices among general women included the use of contraceptives (mainly hormonal pills and condoms) and self-management of menstrual pain and cramps. SRH self-care practices among pregnant women were self-monitoring of ANC, self-medication for pain, self-awareness of healthy habits and medical abortion (tele-abortion). A few also reported practicing self-care in the form of pre-exposure prophylaxis for HIV (self-medication), using condoms and being aware of self-testing for HIV.

There was a need for general as well as need-based information (e.g., telephone counselling on self-management of menstrual cramps, symptoms management, and information on pregnancy). Psychosocial support while seeking medical abortion was felt particularly relevant for pregnant women and postpartum women. A client-centered approach among service providers, as well as the provision of confidential HIV testing, were some of the additional needs expressed.

## Barriers of adopting self-care practices for SRH

The barriers of adopting selfcare practices for SRH includes poor knowledge regarding SRH self-care practices, limited access to SRH self-care services, negative attitude of service providers and program related barriers.

**Poor knowledge and perception regarding SRH self-care practices.** Participants mentioned that most women in the community did not have information about contraceptive devices and used hormonal pills as birth control without a prescription. A participant explained that she did not need to make postnatal visits to see the doctor after childbirth.

When asked about self-testing for HIV, one of the participants had previously performed the test. The rest of the participants had mixed opinions as some did not feel the need for it, as they did not do multiple partnering, while some received help while testing for HIV and hepatitis at the health institution. Perceived fear of getting inaccurate test results (e.g., HIV self-care kits or pregnancy tests), fear of self-testing for HIV was also explained as a reason for not doing the tests.

*Once, I performed myself using a pregnancy test kit. It was shown negative by the kit but the actual result was positive at the third time of using it. It was shown positive by the urine test in the hospital. (R1, General women, user of pregnancy test kit)*

The health care provider suggested that misbeliefs surrounding the low sensitivity and reliability of pregnancy test kits and HIV tests were the reason for a lower uptake.

*Self-test kit does not have sensitivity and specificity of 100%, so that it can give false negatives too sometimes. (Health Care provider)*

Most women had used a pregnancy test kit at least once but believed that the pregnancy kit is not 100% accurate and there was a need for the doctors to confirm their pregnancy.

*I tested with the pregnancy kit and we shared the result of pregnancy test kit with the family. However, my family were not convinced of the pregnancy test kit result and I was suggested to go to the hospital. I went to the hospital and the doctor asked me to use the pregnancy kit. This time my family members were convinced of the result. (Pregnant women)*

All the participants agreed that they did not feel the need for SRH self-care at first and were hesitant in using self-care tools, such as contraceptives and self-testing. Sometimes they were also advised by their family members or friends not to use the tools, as the women 'could not decide themselves'. A participant mentioned that she believed it when they said that using a contraceptive device before having the first child causes infertility. Limited information about selfcare, women's inability to make autonomous decisions and dependence on family members and service providers for information and services was the reasons for lower uptake.

*Still, we hesitate to use contraceptives. My parents have warned me not to use them unless I have my first child. We don't know if the provided information is valid and reliable or not. (Pregnant women)*

**Limited access to SRH services.** The service users and service providers believed that, besides low awareness, limited access of SRHR information and services was the reason why self-care practices had low uptake rates among the target groups. They highlighted the increased need for the availability and accessibility of self-care-related information and services. In line to this, here is what a participant shared:

*The reason for me not using the self-care measures beyond the pregnancy kit and pregnancy mobile application is the lack of information and awareness regarding other selfcare measures. (Pregnant women)*

**Negative attitude and behaviors of service providers.** One woman who sought safe abortion services in the past mentioned being lectured by the health service providers not to abort, which increased her guilt and mental stress about that decision. She speculated that this could be the case for others who may seek medical abortion services.

*When my husband and I decided to have an abortion, I felt so bad when the doctor treated me in that way. I get emotional when I remember that incident as if I am guilty about it. The doctor insisted that I continue my second pregnancy too, since I already had my first baby. I had no interest in continuing the pregnancy because of the stress while having my first baby. (General women seeking abortion service)*

The women did not like having to be overly concerned about the pregnancy. Women prefer self-management of pregnancy through mobile apps.

*I feel sometimes doctors are too concerned for us than me myself. The doctors constantly talk about the need to gain weight, gestational weight etc. which is at times irritating and frustrating. (Pregnant women)*

**Program-related barriers.** The healthcare providers and community workers emphasized the lack of evidence as the reason why it was so difficult to assess SRH self-care practices such as PrEP intake or HIV self-testing. Limited evidence on effectiveness and efficacy of self-care methods, as well as low quality of care could be the reasons for failing to address the needs of target populations, leading to lower uptake rates. The lack of evidence is a gap on the target

groups' needs and hindered program managers' abilities to develop tailored information on self-care practices.

*The scale up of PrEP services is obviously not as they should have been but if we compare PrEP piloting back then and the situation now, beneficiaries are enjoying the PrEP services. There are still issues of quality and continuation among the vulnerable population. We do not have research on whether the reason is due to a decrease in sexual practices, quality of services or any other factors. (Community workers working on HIV and AIDS)*

According to the service providers, national SRH policies and guidelines are not able to address all the standards and dimensions of the services recommended. For instance, limiting the provision of medical abortion services to only selected (trained) service providers could be one of the barriers, as not all women have access to those health service providers. There was also an increased need for disseminating the information contained in policy and guidelines to the grassroots level, in order to enable the target groups to seek services and information. Participant R9 asks questions:

*The message needs to spread in the community because we're talking about self-care in the Interim Guideline but how will they seek the services if community people do not know about it? How will they consult with the providers? (Health care provider)*

## Facilitators of adopting selfcare practices for SRH

Facilitators that improved SRH self-care practices included: information and knowledge regarding selfcare practices, peer support, increasing the number of NGO-based service sites, and use of digital health tools.

**Information and knowledge regarding self-care practices.** Regular consultations by health care providers and community workers with women and girls, promoting SRH self-care interventions and adequate information regarding the SRH self-care interventions were perceived as helpful in understanding the need for self-care related to SRH and increased SRH self-care service utilization.

*I know about the clinic where a self-test kit was provided. I wanted to test my HIV status and also wanted to know the process of self-testing. Thus, I went to clinic to self-test my HIV and AIDS. There are supervised self-testing where service provider guide, unsupervised where a leaflet is provided guiding the beneficiary for HIV-self testing and Dispensing HIV self-test kit where beneficiaries can take away the HIV test kit. (Pregnant women)*

The health care provider emphasized that counselling for participants to help use HIV testing kit, ART services, medical abortion services etc. also had positive effects among women.

*Self-test kits are reliable, but they are used independently so we doubt how they perceive the result. We are unsure if they interpret the result well and they accept the result of the HIV self-test kit or not. Though they use a self-test kit, we still follow up with them. Client education is important for self-testing and for the ART services as well. (Health care provider)*

**Family and peer support.** Women explained the benefits of SRH self-care for themselves. Further, they mentioned advising their family and relatives to adopt self-care practices,

particularly to those who were pregnant and lactating due to their tendency to follow restrictive traditional practices. Participant R4 explained that women undergo many changes in various phases of the lifecycle and urged that other women in her network were in need and would benefit more from self-care practices. The participants agreed that not just support from peers, but support from family (such as husband) and friends would also be important to encourage self-care practices.

*One of my friends suggested me to use the pregnancy app that I have been using currently. It helps you track your pregnancy, provides an overview of what to expect and some graphic pictures as well which is interesting. At times, I also ask friends who have already been mothers to get answers for practical pregnancy related queries. (Pregnant women)*

**Increasing number of NGO-based service sites.**   The number of community organizations working on promoting self-care and SRH services has been increasing, as mentioned by the women and community workers. Most participants also mentioned that access to self-care services and information had been increasing. Some NGO-based service centers were working to make HIV self-testing kits available for those in need, which helped the target groups access self-testing kits as well as also increasing awareness about prevention.

*There are several service centers they provide HIV testing services, and they share information about various available options. There is a self-testing kit that is performed in the presence of friends in the community who are trained in this. The other way is to teach them the methodology and process so they can perform self-testing on their own. (Community worker)*

**Use of digital (health) tools.**   A growing trend towards calling toll-free numbers or using an app was evident among participants, as it would provide timely information instead of visiting the hospital and eliminate the possibility of facing discrimination in healthcare settings, particularly for women with disabilities. Most of the participants used mobile phone-based applications, such as Google, as a source of information and for finding out about the services, and although they were aware of a toll-free number, they barely used it. Curiosity about the health and conditions of the baby was the reason for acquiring more information online during pregnancy.

*HA have simple health knowledge but may be not especially on women's related health problems. I call on Toll free number of an organization especially working women's sexual and reproductive health issues if I get any problems. It helps a lot particularly for middle class family like us, we can ask as many queries as we want. (Women with disability)*

The majority of the participants tracked their menstrual period dates and managed their health visit plans during pregnancy with the help of mobile phone-based apps. Participants were aware of young people purchasing various self-care-related reproductive health products online for privacy and comfort. From the program perspective, digital media platforms and social media platforms helped organizations and program managers disseminate and assess information regarding self-care.

*There is also a trend to purchasing different RH products online. It is because privacy is ensured while doing so. People feel more comfortable buying them online than going in a shop*

*and asking in front of all. This has increased people's interest toward the online market these days. (Community worker)*

The participants generally discussed how social media could be an effective communication tool that can reach wider target groups with self-care-related information, as it is easily accessible, faster and has a wide coverage. Moreover, it could replace the traditional face-to-face consultation, and with that, information about tests and associated costs can be reduced, which further enables people to make informed decisions about health care,

## Discussion

To our knowledge, this is the first study that explored the barriers and facilitators to practice SRH self-care among the target groups of SRH interventions in the Nepalese context. The commonly practiced self-care among our study participants were self-administration of contraceptives, self-management of pain, self-monitoring of pregnancy, self-awareness and seeking medical abortions (tele-abortion), self-medication of pre-exposure prophylaxis for HIV, and self-testing for HIV and pregnancy. The barriers that could hinder practicing self-care were poor knowledge and lack of perception of need, limited access, stigma and negative attitude of service providers, and program-related barriers. While peer support, an increasing number of service sites, and the use of digital tools were the facilitators.

Of note, the high level of awareness and willingness to practice self-care reported by our study participants is partially because of the highly selective, educated sample of the individuals that were within the reach of the SRH interventions. In fact, we learned from our participants that self-care practices for SRH are generally low among various target groups. There is an urgent need to address the issue of poor knowledge of self-care and SRHR interventions to increase access to SRH self-care, thereby achieving global development goals and commitments [7]. Population-wide awareness programs supplemented by increasing service sites are essential for increasing SRH self-care practices. Multi-level interventions, including a safe and supportive enabling environment such as availability of services, awareness regarding self-care measures, and prioritized self-care services, are essential [1].

Our participants seemed very thoughtful about helping their family and friends, which indicates that a peer-based approach might facilitate the adoption of SRH self-care practices in the community. Murray et al. suggested that self-care practices not only lead to improved health conditions but could also be a tool for empowerment and for helping significant others through knowledge sharing [19].

The benefits offered by digital tools within the realm of a multiplicity of SRH-self-care practices, as also suggested by previous studies conducted in various settings, provide the prospects for integrating digital tools into mainstream SRH service systems [5,20–22]. Self-care interventions are found to be utilized and accepted more if they are offered through digital tools [20–22]. Moreover, digital platforms are perceived to be less influenced by barriers, such as social stigma, and are perceived by the target groups to be safe and increase their self-efficacy [23].

Apart from the barriers at the individual level, such as low SRH literacy and poor access, we also noted some important program and policy-related barriers, including health service providers' negative attitudes and poor knowledge, which jointly contributed to the inaccessibility of SRH-related services. The suggested multilevel barriers demand a multilevel approach towards developing interventions to promote the practice of self-care. Interventions to increase general awareness about SRH self-care tools, including strategies to address stigma and intersecting forms of discrimination, and increase tolerance towards women from all backgrounds, are necessary. Community-based approaches, such as using peer-based

approaches, could also challenge existing societal norms and taboos attached to SRH-related issues, creating an enabling environment to practice and make SRH self-care-related informed choices [24].

The Right to Safe Motherhood and Reproductive Health Act of 2018 has clearly stated the right of women to access safe abortion services without coercion and discrimination. However, the present study found that health care providers did not do well while facilitating this decision but rather influenced patients to dismiss the idea of safe abortion. Any counselling sessions in relation to SRHR should be respectful towards patients' backgrounds, and ensure autonomy in the decisions, privacy, and confidentiality, regardless of patients' (health) literacy levels [25]. Access to SRH self-care information and tools can also increase trust among the health service providers, a report by the Asia Pacific Alliance for Sexual and Reproductive Health and Rights suggests [3].

Furthermore, the current SRH programs should also have strategies to strengthen the implementation by ensuring the availability and expansion of self-testing kits at the community level, lowering the out-of-pocket expenditure related to SRH services and changing service providers' negative attitudes. This would directly facilitate access to self-care interventions, as also indicated by the experience of Uganda and Nigeria in addressing all the barriers to increasing universal access to SRH services [26].

While addressing the issues of increasing access, higher program costs could be an issue. It may be necessary to spool funds through public-private partnership on cost-effective self-care tools, and through increasing social insurance coverage [8]. However, there is a general lack of evidence on how we could reduce the total cost of self-care interventions, which demands future research on the topic [3,27].

## Study limitations

Our study has some limitations. First, we have a very small purposive sample, which may not have been enough to generate data saturation, and the participants recruited do not represent all members of the population segment studied. We did not recruit participants of various education levels and cultural backgrounds, which could have impacted the richness of the data. However, having a highly educated sample of participants who could better explain the barriers and facilitators of self-care would be relevant for exploring an under-studied topic.

Second, the interview guide used in this study was pilot tested among two participants only for face validation but not for content and construct validation. Third, the findings could have been partially influenced by researchers' subjectivity. However, the first author, who conducted and moderated all the interviews, is well-trained and skilled in conducting qualitative interviews, who could potentially respect neutrality and control personal bias and expectations. We also attempted to decrease the limitations through maximizing the responses by interviewing health service providers and community workers, which helped to triangulate the information from different backgrounds of participants.

Finally, due to a very small sample size, our study may not have adequately reflected all the insights on cultural and contextual barriers impeding SRH self-care practices. Since the scope of this study was purely exploratory, the applied qualitative method can be believed to generate enough information to understand the phenomenon under study and provide a relevant basis for future studies in this area and for future interventions.

## Conclusions

Despite the global efforts to promote SRH self-care practices, the provision of SRH services and the uptake are hindered by various underlying barriers such as limited access, negative attitudes

of service providers, and program-related obstacles. While there has been the development of new service sites and digital health interventions, there is a need for additional community-based awareness-raising programs, including peer-based approaches, and expansion of sites providing SRH self-care services to engage diverse target groups and promote and enable informed decision-making regarding SRH self-care. LMICs such as Nepal must be a priority for promoting women's access to SRH self-care practices at the regional and global levels.

## Supporting information

**S1 File.**
(PDF)

## Author Contributions

**Conceptualization:** Amit Timilsina, Buna Bhandari, Alexandra Johns, Subash Thapa.

**Data curation:** Amit Timilsina.

**Formal analysis:** Amit Timilsina, Buna Bhandari, Alexandra Johns, Subash Thapa.

**Investigation:** Amit Timilsina.

**Methodology:** Amit Timilsina, Buna Bhandari, Subash Thapa.

**Project administration:** Amit Timilsina.

**Software:** Amit Timilsina.

**Supervision:** Buna Bhandari, Subash Thapa.

**Validation:** Amit Timilsina.

**Writing – original draft:** Amit Timilsina, Buna Bhandari, Alexandra Johns, Subash Thapa.

**Writing – review & editing:** Amit Timilsina, Buna Bhandari, Alexandra Johns, Subash Thapa.

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
