## [Decision Letter · Decision Letter 0]

29 Feb 2024

PONE-D-23-31576Barriers and Facilitators to Self-Care Practices for Sexual and Reproductive Health in Nepalese women: Insights from users and service providersPLOS ONE

Dear Dr. Timilsina,

Thank you for submitting your manuscript to PLOS ONE. After careful consideration, we feel that it has merit but does not fully meet PLOS ONE’s publication criteria as it currently stands. Therefore, we invite you to submit a revised version of the manuscript that addresses the points raised during the review process.

We look forward to receiving your revised manuscript.

Kind regards,

Oluwatosin Oluwaseun Olu-Abiodun

Academic Editor

PLOS ONE

Journal Requirements:

3. We note that your Data Availability Statement is currently as follows: All relevant data are within the manuscript and its supporting informations.

Reviewers' comments:

Reviewer's Responses to Questions

**Comments to the Author**

1. Is the manuscript technically sound, and do the data support the conclusions?

Reviewer #1: Yes

Reviewer #2: Yes

2. Has the statistical analysis been performed appropriately and rigorously? 

Reviewer #1: Yes

Reviewer #2: Yes

3. Have the authors made all data underlying the findings in their manuscript fully available?

Reviewer #1: Yes

Reviewer #2: Yes

4. Is the manuscript presented in an intelligible fashion and written in standard English?

Reviewer #1: Yes

Reviewer #2: Yes

5. Review Comments to the Author

Reviewer #1: The manuscript was presented with a technically sound piece of scientific research backed with data. The data provided supports the conclusions. The conclusions drawn appropriately based on the data presented.The language used for submission was clear.

Reviewer #2: The manuscript is technically sound, and the data supports the conclusion. However, the conclusion could be improved to reflect/capture all sub-themes that emerged in the study according to their main theme (SRH self-care practices, facilitators, and barriers).

Discussion: all sub-themes could be discussed

Line 221: Limited access should be separated from poor knowledge as a sub-heading for discussion.

Line 107: self-care practices or experiences. Which variable are the authors studying? The word should be operationalized for clarity.

Data from service providers was not reported.

This manuscript was prepared in an intelligent fashion and standard English.

6. PLOS authors have the option to publish the peer review history of their article (what does this mean?). If published, this will include your full peer review and any attached files.

Reviewer #1: **Yes: **Dr. Jeminat Omotade Sodimu

Reviewer #2: **Yes: **ABARIBE, Chidinma Emeka

---

## [Author Response · Author response to Decision Letter 0]

17 Apr 2024

21-03-2024

Oluwatosin Oluwaseun Olu-Abiodun

RE: Revision of Manuscript B-4942-Manuscript-33258

Dear Editor,

We would like to thank you and the reviewers for your time and for providing constructive comments on our manuscript entitled "Barriers and facilitators to self-care practices for sexual and reproductive health among women of reproductive age."

A revised version (clean and track changes) reflecting the point-by-point response to the reviewers’ comments has been submitted for your consideration. We believe we have addressed all the comments and the manuscript has significantly improved and ready for publication in your esteemed journal.

Should you require any additional information, please do not hesitate to contact me. Thank you very much for your time and kind consideration. 

We look forward to getting our manuscript published soon. 

Your sincerely,

Amit Timilsina

RESPONSE TO THE REVIEWERS

Journal Requirements:

RESPONSE: The style requirement of Plos One has been reviewed and all necessary adjustments have been made. 

RESPONSE: The raw data are context and issue sensitive thus to respect the confidentiality and anonymity of respondent, the raw data cannot be shared open. However, if the data is required for anyone, it can be made available upon direct request to the first author. 

3. We note that your Data Availability Statement is currently as follows: All relevant data are within the manuscript and its supporting informations.

RESPONSE: The codebook produced as the outcome of the data analysis, the interview guideline for the IDI and KII, and the informed consent forms have been available for this study. As participants have not consented to the availability of full interview transcripts during this study, it has not publicly available to respect participants' anonymity and consent.

RESPONSE: This section has been updated in Methodology ( Line: 134-136)

Changes: The Nepal Health Research Council (Ref. No: 3093) reviewed the research proposal provided ethical clearance for the study, and approved the study. The research follows standard ethical principles guided by the Helsinki Ethics Principles. The autonomy and confidentiality of participants have been maintained throughout the project period. All the participants were oriented about the study and its importance. Before data collection, informed written consent was taken from the participants after orientation, verbal consent was taken before starting the interviews and their right to withdraw from the research was reiterated. The participants were assured of anonymity and confidentiality before, during and after the study. 

RESPONSE: We have double-checked the reference list, edited and ensured that it is complete and correct. 

Reviewers' comments:

5. Review Comments to the Author

Reviewer #1: The manuscript was presented with a technically sound piece of scientific research backed with data. The data provided supports the conclusions. The conclusions drawn appropriately based on the data presented. The language used for submission was clear.

RESPONSE: We would like to thank the reviewer for appreciating our work. 

Reviewer #2: The manuscript is technically sound, and the data supports the conclusion. However, the conclusion could be improved to reflect/capture all sub-themes that emerged in the study according to their main theme (SRH self-care practices, facilitators, and barriers).

RESPONSE: Thank you for your suggestion. The revised conclusion in the abstract and the main text include the study's key findings/themes. 

Discussion: all sub-themes could be discussed

Line 221: Limited access should be separated from poor knowledge as a sub-heading for discussion.

RESPONSE: We have revised the discussion section and tried to include the most relevant sub-themes. The sub-heading “limited access” has been separated according to reviewer’s suggestion. 

Line 107: self-care practices or experiences. Which variable are the authors studying? The word should be operationalized for clarity.

Data from service providers was not reported. (Line 328 reports)

RESPONSE: Our interest is in self-care practices and operationalized in the methods. For consistency, we have used the phrase, self-care practices, throughout the main text. 

Page no. Lines 290, 357, and 379 include data from the service providers (health care providers), while lines 344 and 405 provide data for the community workers. The service providers mentioned in this study include both healthcare providers and community health workers.

---

## [Editor Report · Decision Letter 1]

6 May 2024

Barriers and facilitators to self-care practices for sexual and reproductive health among women of reproductive age.

PONE-D-23-31576R1

Dear Amit Timilsina,

We’re pleased to inform you that your manuscript has been judged scientifically suitable for publication and will be formally accepted for publication once it meets all outstanding technical requirements.

Kind regards,

Dr. Oluwatosin Olu-Abiodun

Academic Editor

PLOS ONE

---

## [Editor Report · Acceptance letter]

14 May 2024

PONE-D-23-31576R1 

PLOS ONE

Dear Dr. Timilsina, 

I'm pleased to inform you that your manuscript has been deemed suitable for publication in PLOS ONE. Congratulations! Your manuscript is now being handed over to our production team.

Kind regards, 

on behalf of

Dr. Oluwatosin Oluwaseun Olu-Abiodun 

Academic Editor

PLOS ONE